# Dynamic Regulation of miRNA Expression by Functionally Enhanced Placental Mesenchymal Stem Cells PromotesHepatic Regeneration in a Rat Model with Bile Duct Ligation

**DOI:** 10.3390/ijms20215299

**Published:** 2019-10-24

**Authors:** Jae Yeon Kim, Ji Hye Jun, Soo Young Park, Seong Wook Yang, Si Hyun Bae, Gi Jin Kim

**Affiliations:** 1Department of Biomedical Science, CHA University, Seongnam 13488, Korea; janejykim92@gmail.com (J.Y.K.); jihyejun1015@gmail.com (J.H.J.); soobookpark@gmail.com (S.Y.P.); 2Department of Systems Biology, College of Life Science and Biotechnology, Yonsei University, Seoul 120749, Korea; yangsw@yonsei.ac.kr; 3Department of Internal Medicine, Catholic University Medical College, Seoul 03312, Korea; baesh@catholic.ac.kr

**Keywords:** Liver failure, microRNAs (miRNAs), placenta-derived mesenchymal stem cells (PD-MSCs), phosphatase of regenerating liver-1 (PRL-1), regenerative medicine, stem-cell homing, vascular remodeling

## Abstract

Placenta-derived mesenchymal stem cells (PD-MSCs) were highlighted as therapeutic sources in several degenerative diseases. Recently, microRNAs (miRNAs)were found to mediate one of the therapeutic mechanisms of PD-MSCs in regenerative medicine. To enhance the therapeutic effects of PD-MSCs, we established functionally enhanced PD-MSCs with phosphatase of regenerating liver-1 overexpression (PRL-1(+)). However, the profile and functions of miRNAs induced by PRL-1(+) PD-MSCs in a rat model with hepatic failure prepared by bile duct ligation (BDL) remained unclear. Hence, the objectives of the present study were to analyze the expression of miRNAs and investigate their therapeutic mechanisms for hepatic regeneration via PRL-1(+) in a rat model with BDL. We selected candidate miRNAs based on microarray analysis. Under hypoxic conditions, compared with migrated naïve PD-MSCs, migrated PRL-1(+) PD-MSCs showed improved integrin-dependent migration abilitythrough Ras homolog (RHO) family-targeted miRNA expression (e.g., hsa-miR-30a-5p, 340-5p, and 146a-3p). Moreover, rno-miR-30a-5p and 340-5p regulated engraftment into injured rat liver by transplantedPRL-1(+) PD-MSCs through the integrin family. Additionally, an increase inplatelet-derived growth factor receptor A (PDGFRA) by suppressing rno-miR-27a-3p improved vascular structure in rat liver tissues after PRL-1(+) PD-MSC transplantation. Furthermore, decreased rno-miR-122-5p was significantly correlated with increased proliferation of hepatocytes in liver tissues by PRL-1(+) PD-MSCs byactivating the interleukin-6 (IL-6) signaling pathway through the repression of rno-miR-21-5p. Taken together, these findings improve the understandingof therapeutic mechanisms based on miRNA-mediated stem-cell therapy in liver diseases.

## 1. Introduction

Although the liver has exceptional regenerative ability, hepatic diseases are induced by several environmental factors, such as viral infection and chemical exposure [1]. Accumulated fibrosis, progressive hepatic vascular pressure, and inflammatory reaction due to continuous hepatic damage are the main causes of liver cirrhosis [2]. In particular, failure of the hepatocyte–endothelium crosstalk in the damaged liver results in abnormal healing, which is shown by the formation of fibrosis or scar tissue [3]. Additionally, the abnormal hepatic vasculature system not only represses metabolic ability but also exhibits fibrotic pathophysiology [4,5]. Presently, orthotopic transplantation of the liver is credited asthe only therapeutic surgical method to treat irreversible hepatic failure. However, this operation has many limitations (e.g., insufficiency of donors and the problem of immunity).

Recently, the facilitation of hepatic regeneration in chronic liver injuries by mesenchymal stem cells (MSCs) was shown to be a useful cell therapy source [6,7]. In previous reports, we demonstrated that placenta-derived MSCs (PD-MSCs) have a therapeutic effect on carbon tetrachloride (CCl_4_)-injured rat liver through antifibrotic, antiapoptotic, and autophagic mechanisms and epigenetic alterations of interleukin-6 (IL-6)/signal transducer and activator of transcription 3 (STAT3) signaling [8,9]. Also, the migration activity of MSCs into injured tissues is an important factor in maximizing their therapeutic effects, because transplanted MSCs adhere to the endothelium of vessel and migrate into damaged tissues through chemoattraction [10,11]. We previously demonstrated that PD-MSCs migrated endothelial cells in a hypoxic environment through integrin alpha 4 (ITGA4) and Ras homolog (RHO) signaling [12]. Integrin-dependent signaling activates adhesion to enhance MSC migration into injured target tissues. Additionally, migrated MSCs promote microvessel regeneration in damaged tissues [13]. It is well known that MSCs have angiogenic paracrine effects via secreted molecules, including several growth factors and secretomes, as well as vascular remodeling, which increase the proliferation and viability of endothelial cells. However, the mode of action of naive MSCs is still unclear, as well as the low efficacies of MSCs in degenerative diseases. Due to the reason, many scientists tried developing next-generation stem cells, which have functionally enhanced potentials. 

Recently, it was reported that phosphatase of regenerating liver-1 (PRL-1; protein tyrosine phosphatase type IVA, member 1; PTP4A1; PTPCAAX1), which was identified as an immediate early gene, is involved in vascular regeneration by increasing portal flow in a partial hepatectomy model during hepatic regeneration [14] and in mitogenic upregulation [15]. Moreover, PRL-1 induces migration and adhesion by activating c-Src levels, and p115 Rho GTPase-activating protein (GAP) binds PRL-1 through the Src homology 3 domain [16]. In particular, microRNA (miR)-26a, miR-944, and miR-601 suppressed cell migration and invasion by targeting PRL-1 [17,18,19].

MicroRNAs (miRNAs) are small, non-coding, single-stranded RNA sequences, 18–22 nucleotides in length, whichregulate diverse cellular processes by binding to the 3’untranslated region (3’-UTR) of target messenger RNAs (mRNAs), resulting in mRNA degradation and translational repression [20]. MiRNAs are important regulators of stem cells in the treatment of various diseases, such as liver fibrosis [21]. Downregulation of miR-30e expression in a CCl_4_-induced hepatic fibrosis model by human bone marrow-derived MSCs (hBM-MSCs) resulted in migration ability [22]. Moreover, umbilical cord MSC (UC-MSC) transplantation reduced the severity of hepatic fibrosis in a CCl_4_-injured mouse model. The suppression of miR-199 expression targetedkeratinocyte growth factor mRNA, leading to adecrease in translation [23].

However, the profile and functions of the miRNAsthat mediate liver regeneration followingthe administration of functionally enhanced PD-MSCs with PRL-1 overexpression (PRL-1(+)) to the bile duct ligation (BDL)-injured rat model remainunclear. Therefore, the major objective of the study was to analyzethe expression patterns of miRNAs and toinvestigate miRNA-mediated therapeutic effects on hepatic regeneration usingthe engraftment of PRL-1(+) PD-MSCs, through monitoring changes in adhesion molecules and angiogenic factor-targeted miRNAs in a rat model with BDL.

## 2. Results

### 2.1. MiRNA Profiling of Naïve PD-MSCs under Hypoxic Conditions and in BDL-Injured Liver in Rats

To determine miRNA-mediated regulation of the migration ability of PD-MSCs, we assessed microarray of migrated naïve PD-MSCs under hypoxic conditionsand from theliver samples of rats with BDL administered naïve PD-MSCs at one and twoweeks. Firstly, the mRNA expression profile revealed significantly increased hypoxia-inducible factor 1 alpha (HIF-1α) and vascular endothelial growth factor (VEGF) in the migrated naïve PD-MSCs under hypoxic conditions compared with those under normoxic conditions (Figure 1A, * *p* < 0.05). In the microarray data, 57 miRNAs were detected in migrated naïve PD-MSCs under hypoxic conditions and in BDL-injured rat liver samples transplanted with MSCs.We found thattwo miRNAs were downregulated and 18 miRNAs were upregulated under hypoxic conditions compared with normoxic conditions (Hyp/Nor), in the transplanted naïve (TTx Naïve) group at one week compared with the non-transplanted (NTx) group at one week (TTx/NTx 1w), and in the TTx Naïve group at two weeks compared to the NTx group at two weeks (TTx/NTx 2w). Moreover, we identified threedownregulated miRNAs and 34 upregulated miRNAs shared by the Hyp/Nor and TTx/NTx 1w comparisons and nine downregulated miRNAs and 25 upregulated miRNAs shared by the Hyp/Nor and TTx/NTx 2w comparisons. Also, we identified three down regulated miRNAs and 26 upregulated miRNAs shared by the TTx/NTx 1w and TTx/NTx 2w comparisons (Figure 1B,C). The data suggested that miRNA profiling of naïve PD-MSCs with migration ability under hypoxicconditions and of BDL-injured rat liver samples was confirmed.

### 2.2. PRL-1-Dependent Migration Ability under Hypoxic Conditions Regulatedby miRNAs Targeting the Integrin Family

To investigate whether PD-MSCs with PRL-1(+) regulate adhesion-related molecules for cell migration, we assessed mRNA and targeting miRNA expression in the PD-MSCs with PRL-1(+) under hypoxic conditions. We analyzed the migration ability of naïve PD-MSCs and PD-MSCs with PRL-1(+) using a Transwell insert system under 1% hypoxic conditions. The number of the naïve PD-MSCs under hypoxic conditions was increased compared to normoxia. Interestingly, PRL-1(+) PD-MSCs under hypoxia significantly improved migration ability compared to normoxia and naïve conditions, as shown in Figure 2A,B (^#^
*p* < 0.05). Increased HIF-1α and VEGF levels were confirmed in the PD-MSCs with PRL-1(+) under hypoxic conditions compared with those under normoxic conditions (Figure 2C, * *p* < 0.05). Also, the expression levels of Ras homolog family member A (RHOA) and RHO-associated coiled-coil-containing protein kinase 1 (ROCK1) were significantly increased in migrated PRL-1(+) PD-MSCs compared to migrated naïve PD-MSCs under hypoxic conditions (Figure 2D, ^#^*p* < 0.05). Interestingly, the mRNA expression of PRL-1 was increased in migrated naïve PD-MSCs under hypoxia compared with those under normoxia. Moreover, hypoxia-treated PRL-1(+) PD-MSCs had higher PRL-1 expression than normoxia. We found that the hsa-miR-30a-5p binding site is conserved in the 3’-UTR of the PRL-1 mRNA. Furthermore, hsa-miR-30a-5p expression clearly matched the mRNA expression of PRL-1 in naïve and PRL-1(+) PD-MSCs under hypoxic conditions (Figure 2E, ^#^
*p* < 0.05). The naïve PD-MSCs showed slightly decreased integrin alpha 4 (ITGA4) expression under hypoxic conditions compared with normoxic conditions. The evidence supported our previous report that decreased ITGA4 expression controls the migration ability of hypoxic naïve PD-MSCs [12]. On the other hand, migrated PRL-1(+) PD-MSCs exposed to hypoxia had a significant increase in ITGA4 expression. The targeting of hsa-miR-340-5p to ITGA4 was confirmed (Figure 2F, ^#^
*p* < 0.05). Additionally, integrin beta 7 (ITGB7) expression in migrated naïve PD-MSCs was higher under hypoxia than under normoxia. Interestingly, we found that migratedPRL-1(+) PD-MSCs under hypoxia had remarkably increased ITGB7mRNA levels. Moreover, hsa-miR-146a-3p expression was capable of more markedly downregulating the expression of ITGB7 in PRL-1(+) PD-MSCs than in naïve (Figure 2G, ^#^
*p* < 0.05). Also, we further analyzed if hsa-miR-30a-5p, 340-5p, and 146a-3p directly target PRL-1, ITGA4, and ITGB7 in naïve and PRL-1(+) PD-MSCs using a luciferase assay (Figure 2H,I, ** *p* < 0.05). Overexpression of miR-30a-5p, 340-5p, and 146a-3p in the PRL-1(+) under hypoxic conditions using an miRNA mimic decreased miRNA expression compared to the naïve, whereas mRNA expression of each target gene was increased (Appendix A). These results suggest that the hypoxia-mediated migration ability of PRL-1(+) PD-MSCs regulates miRNAs through the integrin family.

### 2.3. PRL-1-Targeted miRNA Expression Regulates Migration Ability through the RHO Family

Interestingly, we found that PRL-1(+) PD-MSCs had increased migration ability under hypoxic conditions compared with normoxic conditions. Therefore, we hypothesized that PRL-1 may be positively related to migration ability. To confirm the function of PRL-1 in the migration of PD-MSCs, we used small interfering RNA (siRNA) PRL-1 (siPRL-1) treatment to knockdown PRL-1 expression. The number of the PRL-1(+) PD-MSCs was higher than that of naïve. On the other hand, the migration ofnaïve and PRL-1(+) PD-MSCs significantly decreased with siPRL-1 treatment (Figure 3A,B, ^#^
*p* < 0.05). Moreover, we verified PRL-1 expression with or without siPRL-1 treatment. hsa-miR-30a-5p expression was decreased in the PRL-1(+) PD-MSCs compared to the naïve PD-MSCs. The knockdown of PRL-1 induced increased hsa-miR-30a-5p levels (Figure 3C, ^#^
*p* < 0.05). The expression levels of RHOA and ROCK1 were significantly increased in the PRL-1(+) PD-MSCs and without siPRL-1 treatment compared to the naïve PD-MSCs. Following siPRL-1 treatment, the mRNA expression levels of RHOA and ROCK1 were clearly attenuated in the naïve and PRL-1(+) PD-MSCs (Figure 3D, ^#^
*p* < 0.05). In addition, protein levels of RHOA and ROCK1 in the PRL-1(+) PD-MSCs were higher than those of naïve. According to knockdown of PRL-1, RHOA and ROCK1 levels were downregulated (Figure 3E,F, ^#^
*p* < 0.05). The data demonstrate that PRL-1-dependent miR-30a-5p regulates migration through the RHO family.

### 2.4. miRNA Expression Regulates Integrin Family for PRL-1(+) PD-MSC Homing In Vivo in a Rat Model with BDL

One function of MSCs is efficient stem-cell homing and migration into injured target tissue for therapy [24]. Activated integrin family, RHOA, and downstream factor ROCK regulate MSC adhesion and migration by regulating phosphorylated focal adhesion kinase (FAK) [25]. Therefore, we confirmed that the integrin family regulates miRNA expression for stem-cell engraftment in a BDL-injured rat model. After each MSC transplantation, PKH67-positive (green) signals in a frozen section of BDL-injured rat liver was found. Compared to the tail-vein transplantation (TTx) Naïve group, the percentage of PKH67+ signals in the TTx PRL-1(+) group was increased (Figure 4B, **p*<0.05). The human-specific Alu sequence in cirrhotic liver samples from each rat group was confirmed using quantitative real-time polymerase chain reaction (qRT-PCR) analysis. Compared to the non-transplantation (NTx) group, the TTx Naïve group had increased human-specific Alu expression. Interestingly, we found that, compared to the TTx Naïve group, the TTx PRL-1(+) group exhibited a remarkable increase in Alu level (Figure 4B, ^#^
*p* < 0.05). Consistently, ITGA4 and ITGB7 expression in the TTx PRL-1(+) group was significantly increased compared to the TTx Naïve group. We confirmed that the protein expression of ITGA4 and ITGB7 in the TTx PRL-1(+) group was increased compared to the NTx and naive groups (Figure 4C, ^#^
*p* < 0.05). We searched for differences in hsa-miR-30a-5p and rno-miR-30a-5p. Whereas hsa-miR-30a-5p targets PRL-1 mRNA, rno-miR-30a-5p targets integrin alpha 6 (ITGA6) in miRNA-target prediction databases (http://www.mirdb.org and http://www.targetscan.org). Also, rno-miR-340-5p targets integrin beta 1 (ITGB1). The mRNA level of ITGA6 was decreased in the NTx group compared to the TTx group. rno-miR-30a-5p was increased in the NTx groups compared with the TTx groups except at one week. Interestingly, ITGA6 expressionwas significantly increased in the TTx PRL-1(+) group compared with the TTx Naïve group by repressing rno-miR-30a-5p level (Figure 4D, ^#^
*p* < 0.05). Moreover, the ITGB1 expression of the TTx PRL-1(+) group was remarkably improved compared to that of the NTx and TTx Naïve groups by suppressing rno-miR-340-5p level (Figure 4E, ^#^
*p* < 0.05). The results suggested that the miRNA expression regulates theintegrin family for engraftment and PRL-1(+) PD-MSC transplantation in a rat liver model injured with BDL.

### 2.5. Improved Vascular Remodeling by PRL-1(+) PD-MSCs through the Regulation of miRNA Expression by Platelet-Derived Growth Factor Receptor A(PDGFRA) in a BDL-injured Rat Model

Generally, vascular structures and itsfunction in tissues are important factors for organ homeostasis. However, abnormal conditions arise when they are exposed to stress or damaged conditions. Thus, abnormal vascular structures in damaged liver tissues are common. To analyze vascular remodeling following the transplantation of naïve and PRL-1(+) PD-MSCsinto a BDL rat model, we confirmed the expression and localization of angiogenic factors. Compared to that in the NTx group, the mRNA expression of endoglin (ENG) in the TTx Naïve group was increased at two and threeweeks. Interestingly, compared with the TTx Naïve groups, the TTx PRL-1(+) group had significantly improved ENG expression and platelet-derived growth factor receptor beta (PDGFRB) level (Figure 5A,B, ^#^
*p* < 0.05). Furthermore, the PDGF receptor alpha (PDGFRA)-targeted rno-miR-27a-3p was significantly repressed in the TTx Naïve group at two, three, and five weeks and the TTx PRL-1(+) group at one, two, and three weeks compared to the NTx group. In contrast to the expression of rno-miR-27a-3p, the expression pattern of PDGFRA in the TTx PRL-1(+) group was remarkably increased (Figure 5C, ^#^
*p* < 0.05). To investigate the localization and expression of PDGFRA in BDL rat liver tissues transplanted withPRL-1(+) PD-MSCs, we performed immunofluorescence assays. PDGFRA was localized in the membrane of liver sinusoidal endothelial cells and inthe hepatic nucleus. In particular, the expression of PDGFRA was significantly upregulated in the TTx PRL-1(+) group compared with the NTx and TTx Naïve groups (Figure 5D,E,^#^
*p* < 0.05). These results indicate that vascular remodeling is improved by PRL-1(+) PD-MSCsthrough the PDGFRA-mediated regulation of miRNA expression in a rat model of BDL.

### 2.6. miRNAs Mediate Hepatic Regeneration by PRL-1(+) PD-MSCs in a Rat Model with BDL through IL-6/STAT3 Signaling

To determine whether the administration of PRL-1(+) PD-MSCs could induce liver regeneration by regulating miRNAs, interleukin-6 (IL-6)/signal transducer and activator of transcription 3 (STAT3) signaling, which is a representative pathway that promotes liver regeneration and which is a well-known cytokine of hepatocyte protection, the expression of mRNA, protein, and regulating miRNAs was confirmed in rat livers with a BDL-injuredmodel. We examined whether PRL-1(+) PD-MSCs promoted hepatocyte proliferation in rat BDL-injured livers, and proliferating cell nuclear antigen (PCNA) immunohistochemistry was used to analyze liver tissues. The mRNA expression of interleukin-6 receptor (IL-6R) was increased in the TTx Naïve group compared with the NTx group at two and five weeks. Interestingly, compared with the TTx Naïve groups, the TTx PRL-1(+) groups had significantly improved IL-6R levels. rno-miR-21-5p-targeted IL-6R expression was downregulated in the TTx Naïve group at two, three, and five weeks and in the TTx PRL-1(+) group at one and two weeks compared with the NTx group (Figure 6A, ^#^
*p* < 0.05). We analyzed the protein levels of IL-6 and glycoprotein 130 (gp130), which is a type I cytokine receptor of IL-6, and the phosphorylation level of STAT3 in rat BDL-injured liver. Although the gp130 level showed no significant differences among the groups, except forin the sham control (Con) group, the expression of endogenous IL-6 and phosphorylated STAT3 in the liver was higher in the TTx Naïve group than in the NTx group at one week. In particular, the TTx PRL-1(+) group had equally improved IL-6 and phosphorylated STAT3 levels compared with the TTx Naïve group (Figure 6B). Furthermore, to analyze the transcription factors involved in the liver regenerationof rats with BDL-injured liver, the mRNA expression levels of hepatocyte nuclear factor 1 (HNF1) homeobox A (HNF1A) and hepatocyte nuclear factor 4 alpha (HNF4A) were measured and were found to be increased in the TTx Naïve group compared with the NTx group. Prominent increases in HNF1A and HNF4A expressions were observed in the TTx PRL-1(+) group compared to the TTx Naïve group. rno-miR-122-5p, which is a representative liver-enriched miRNA that targets HNF1A in the TTx groups, also caused downregulated expression in the TTx groups compared with the NTx group. In particular, compared with the TTx Naïvegroup, the TTx PRL-1(+) group showed remarkably decreased expression of rno-miR-122-5p (Figure 6C,D, ^#^
*p* < 0.05).To confirm the proliferation of hepatocytesfollowing the transplantation ofPRL-1(+)PD-MSCs, we analyzed immunohistochemical staining for PCNA in rat liver tissues. The largest number of PCNA-positive hepatocytes was observed in the TTx PRL-1(+) group, followed by the TTx Naïve and NTx groups (Figure 6E,F, ^#^
*p* < 0.05).These data indicate that PRL-1(+)PD-MSCs may regulate miRNA-mediated hepatic regeneration through IL-6/STAT3 signaling.

## 3. Discussion

Mesenchymal stem cells (MSCs) have promising potentialin regenerative medicine, due to theirself-renewal, differentiation, and immunomodulatory effects [26,27]. Recent studies revealed that the modulation of miRNA by MSCs is involved in the therapeutic effect between MSCs and injured tissues [28]. In our study, miRNA candidates for stem-cell engraftment and vascular remodeling were selected because they were involved in invaded placenta-derived MSCs (PD-MSCs) under hypoxic conditions, as well as in bile duct ligation (BDL)-injured rat livers at one and two weeks post-transplantation, where they demonstrated effects on liver regeneration (Figure 1). Firstly, integrin-dependent targeting miRNAs (e.g., hsa-miR-30a-5p, has-miR-340-5p, and has-miR-146a-3p) were selected. Target genes were searched in miRNA databases (http://www.mirdb.org and http://www.targetscan.org).

Generally, the migration of MSCs under low oxygen concentrations is affected by altered integrin expression and cell-to-cell adhesion [29]. Our previous reports confirmed that migrated naïve PD-MSCs under hypoxia showed decreased integrin alpha 4 (ITGA4) and increased integrin beta 7 (ITGB7) expression for homing effects through the RHO family [12]. In bone marrow-derived MSCs (BM-MSCs), hypoxia inducible factor 1 alpha (HIF-1α) induced microenvironment factors, including hypoxia and ITGA4 expression, impacting the migration ability of BM-MSCs [30]. We confirmed that ITGA4 expression under hypoxia was slightly decreased in migrated naïve PD-MSCs by suppressing hsa-miR-340-5p. Interestingly, compared with naïve, migrated PRL-1(+) PD-MSCs under hypoxia significantly increased both ITGA4 and ITGB7 expression (Figure 2). These data are well matched with the characteristics of PRLs (e.g., PRL-1, PRL-2, and PRL-3), which identically promote cell migration and invasion [31,32,33]. In general, integrin-mediated adhesion initiates signal transduction by inducing the autophosphorylation of FAK [34,35]. Previous evidence suggested that PRL-1 may regulate the activation of the integrin family. Interestingly, we found that hsa-miR-30a-5p, which targets PRL-1, also regulated ITGA4 andits targeted sequences (Appendix A, Appendix A). Also, hsa-miR-30a-5p regulates PRL-1 and ITGA4 expression in migrated PRL-1+ PD-MSCs under hypoxic conditions. In addition, we confirmed that the knockdown of PRL-1 decreased migration ability and suppressed hsa-miR-30a-5p though the RHO family (Figure 3). Also, Ma et al. suggested that the targeting of ITGA4 by the hsa-miR-30s family decreased the proliferation of human coronary artery endothelial cells.

Therefore, we hypothesized that PRL-1 (+) PD-MSCs in a liver failure model would improve engraftment into targeted injured tissues and mediate repair through MSC migration. To verify this hypothesis, the superior signals of PKH67+ labeled with PRL-1(+) PD-MSCs in BDL-injured rat liver tissues at one week were proven by immunofluorescence compared to that of naïve cells. An increase inthe human-specific Alu sequence was confirmed after the administration of PRL-1(+) PD-MSCs to a rat model of BDL but not after the administration of naïve PD-MSCs. Also, ITGA4 and ITGB7 expression significantly increased in TTx groups. Drescher et al. suggested that cell migration mediated by theadhesion molecule ITGB7 was involved in the outcome of non-alcoholic steatohepatitis [36]. We confirmed that the targeting of integrin alpha 6 (ITGA6), but not ITGA4, by rno-miR-30a-5p and the targeting ofITGB1, but not ITGA4, by rno-miR-340-5p were significantly repressed following transplantation with PRL-1 (+) PD-MSCs (Figure 4). Human MSC engraftment into the CCl_4_-injured liver of a murine model regulated ITGB1 ina cluster of differentiation 44 (CD44)-dependent manner [37]. These results are similar to our data.

The transplanted MSCs were found to undergo endothelial transmigration along sinusoidal endothelial cells [38]. A previous study suggested that increased platelet-derived growth factor (PDGF) levels induced liver regeneration by releasing umbilical cord-derived MSCs (UC-MSCs) in a carbon tetrachloride (CCl_4_)-injured rat model [39]. Moreover, the direct targeting of PDGF receptor beta (PDGFRB) by miR-26b-5p was associated with the negative regulation of angiogenesis and fibrosis in a liver fibrosis model treated with methionine-choline-deficient and high-fat diets [40]. Interestingly, we confirmed that the mRNA expressionlevels of endoglin (ENG), PDGF receptor alpha (PDGFRA), and PDGFRB induced byPRL-1(+) PD-MSC transplantation in a rat model with BDL were higher than that those induced by transplantation with naïve PD-MSC at one week through the suppression of rno-miR-27a-3p and the localization of PDGFRA in the hepatic nucleus and in endothelial cells (Figure 5). In addition, we further confirmed increased CD31, also known as platelet endothelial cell adhesion molecule (PECAM-1), levels by immunofluorescence in the TTx PRL-1(+) group 

We previously reported the hepatic regeneration induced by naïve PD-MSCs involves activated interleukin-6 (IL-6)/signal transducer and activator of transcription 3 (STAT3) signaling and the methylation of inhibited IL-6/STAT3 promoters in a CCl_4_-injured rat model [9]. We confirmed the protein expression of IL-6/STAT3 in the TTx Naïve and the NTx groups of a rat model with BDL. Interestingly, compared with the TTx Naïve group, the TTx PRL-1(+) group had significantly increased IL-6/STAT3 at one, two, and three weeks post-transplantation. This result may suggest that PRL-1 led to the phosphorylation of downstream factors, including STAT3, via Src activation for liver regeneration [41]. Additionally, we found that rno-miR-21-5p-targeted interleukin-6 receptor (IL-6R) expression was significantly repressed in the TTxPRL-1(+) group at one week and two weeks post-transplantation. A previous report indicated that miR-21-5p expression was upregulated in patients with hepatitis B-related acute-on-chronic liver failure compared withcontrols [42]. These results show a possibility of mo-miR-21-5p as a biomarker for the prediction of liver regeneration after stem-cell therapy. Moreover, miR-122, which is a liver-enriched miRNA, plays a central role in liver function and inthe progression of liver disease [43]. In a liver fibrosis model induced by CCl_4_, miR-122-modified adipose-derived MSCs inhibited collagen accumulation by suppressing the activation of hepatic stellate cells (HSCs) [44,45]. Our result indicated that rno-miR-122-5p, whichtargets HNF1A, which is a major transcription factor during liver development and regeneration, was dramatically decreased in PRL-1(+) PD-MSCs compared with naïve (Figure 6), in addition to altered blood chemistry (e.g., aspartate aminotransferase, alanine aminotransferase, total bilirubin, and albumin) in rat serum. However, miRNA expression in rat liver tissues after transplantation withPRL-1(+) PD-MSCswas not the sameeach week because tissue-specific miRNA patterns vary according to the stage of diseases.

In conclusion, our findings provide clear evidence that PRL-1(+) PD-MSCs promote miRNA-mediated MSC migration under hypoxic conditions through integrin-dependent signaling and promote hepatic regeneration via increased engraftment and vascular remodeling. However, our futurestudieswill consider whether specific miRNAs have strict standards for selection and, when transfected into PRL-1(+) PD-MSCs, whether theyrecover hepatic function for liver regeneration. These findings will improvethe understanding of therapeutic mechanisms based on miRNA-mediated stem-cell therapy in liver diseases.

## 4. Materials and Methods

### 4.1. Cell Culture and Gene Transfection

Placentas from healthy women (≥37 gestational weeks) were collected by the Institutional Review Board of CHA Gangnam Medical Center, Seoul, Korea (IRB 07-18). The isolation of naïve PD-MSCs was as previously described, their characterization was confirmed [46,47], and they were maintained in alpha-modified minimal essential medium (α-MEM; HyClone, Logan, UT, USA) supplemented with 10% fetal bovine serum (FBS; Gibco, Carlsbad, CA, USA), 25 ng/mLhuman fibroblast growth factor4 (hFGF4) (PeproTech, Rocky Hill, NJ, USA), 1 µg/mLheparin (Sigma-Aldrich, St. Louis, MO, USA), and 1% penicillin/streptomycin (P/S; Gibco). To overexpress human PRL-1 (phosphatase of regenerating liver-1; protein tyrosine phosphatase type 4 A, member 1; PTP4A1) in naïve PD-MSCs, a PRL-1 plasmid containing the CMV6-AC vector and antibiotic neomycin for mammalswas obtained from Origene(Origene Inc., Rockville, MD, USA). Naïve PD-MSCs (5 × 10^5^ cells/cuvette) were transfected using the 4D AMAXA Nucleofector™ system (Lonza, Basel, Switzerland). After transfection, the cells were maintained in naïve PD-MSC medium containing 1.5 mg/mL neomycin for selection. Cells were maintained below 5% CO_2_ at 37 °C. To induce hypoxia, the cells were placed in a hypoxia chamber and maintained at 1% O_2_ and 37 °C.

### 4.2. Animal Models and MSC Transplantation

All animal experimental procedures were approved by the Institutional Animal Care Committee of CHA University, Bundang, Korea (IACUC-190048). Seven-week-old male Sprague-Dawley (SD) rats (Orient Bio Inc., Seongnam, Korea) were used to induce chronic liver cirrhosis via the common BDL model as previously described [48]. The rats were randomly assigned to each of the following groups: sham control (Con; *n* = 5), BDL-injured nontransplantation (NTx; *n* = 20), naïve PD-MSC transplantation (TTx Naïve; *n* = 20), and PRL-1(+) PD-MSC transplantation (TTx PRL-1(+); *n* = 20). For the administration of each MSC, PKH67 (Sigma-Aldrich)-labeled cells were intravenously transplanted into the tail vein. After one, two, three, and five weeks, rats from each group were sacrificed, and liver tissues were extracted to analyze therapeutic effects using qRT-PCR, Western blotting, and immunostaining.

### 4.3. Quantitative Real-Time Polymerase Chain Reaction (qRT-PCR)

Total RNA was isolated fromnaïve andPRL-1(+) PD-MSCs and rat liver tissues using TRIzol (Invitrogen, Carlsbad, CA, USA). Reverse transcription was performed with 500 ng of total RNA and Superscript III reverse transcriptase (Invitrogen). Complementary DNA (cDNA) was amplified by PCR. In the case of cDNA synthesis for miRNAs, we used the miR-X miRNA First-Strand Synthesis kit (Takara bio, Kusatsu, Shiga, Japan). Real-time PCR was performed using SYBR Master Mix (Roche, Basel, Switzerland) and a CFX Connect™ Real-Time System (Bio-Rad, Hercules, CA, USA). Normalizationwas assessed byhuman and rat glyceraldehyde 3-phosphate dehydrogenase (GAPDH)for gene expression and U6 for miRNA expression. The sequences of the primers are shown in Appendix A. All reactions were performed in at least triplicate.

### 4.4. Immunohistochemistry

To observe the degree of hepatocyte proliferation following transplantation with MSCs or the control, BDL rat liver tissues were stained with anti-PCNA (Santa CruzBiotechnology, Dallas, Texas, USA) using immunohistochemistry. The liver tissues were embedded in paraffin and sectioned. The sectioned tissues were incubated in 3% H_2_O_2_ in methanol to block endogenous peroxidase activity. After antigen retrieval, the tissues were incubated with a primary antibody (1:200) at 4 °C overnight, followed by a 1-h incubation with biotinylated secondary anti-rabbit antibody at room temperature. Incubation with horseradish peroxidase-conjugated streptavidin–biotin complex (DAKO, Santa Clara, CA, USA) and 3,3-diaminobenzidine (EnVision Systems, Santa Clara, CA, USA) was performed to generate a chromatic signal. The samples were counterstained with Mayer’s hematoxylin (DAKO). Additionally, the percentage of hepatocytes with PCNA-positive nuclei relative to the total number of hepatocytes was calculated in randomly selected sections using a digital slide scanner (3DHISTECHLtd., Budapest, Hungary). The experiment was analyzed in at least triplicate.

### 4.5. Immunofluorescence

To confirm hepatic vascular remodeling following the administration of MSCs or the control, the liver tissues from each group (*n* = 5) were sectioned into 7-μm-thick slices and fixed with 4% paraformaldehyde. The tissue sections were blocked using blocking solution (DAKO) for 1 h under dark conditions. The primary antibody against PDGFRA (1:100; Santa CruzBiotechnology) was added in a diluent solution (DAKO) at 4 °C overnight. The secondary antibody, Alexa Fluor™594 goat anti-rabbit immunoglobulin G (IgG) (H+L) (1:250; Invitrogen), was reacted for 1 h. The slides from each group were counterstained with 4’,6-diamidino-2-phenylindole (DAPI) (Invitrogen) and observed by confocal microscopy (LSM 700). Images were analyzed with ZEN blue software (ZEISS). The experiment was performed in at least triplicate.

### 4.6. Western Blotting

Homogenized rat liver tissues were lysed in radioimmunoprecipitation assay (RIPA) buffer (Sigma-Aldrich) supplemented with protease inhibitor cocktail (Roche) and phosphatase inhibitor (Sigma-Aldrich). Briefly, 40 μgof proteinwas separated by sodium dodecyl sulfate polyacrylamide gel electrophoresis (SDS-PAGE). The separated proteins were transferred onto Polyvinylidene fluoride (PVDF) membranes (Bio-Rad). The membranes were incubated with primary antibodies at 4 °C overnight. The following antibodies were used: anti-RHOA (1:1000; Cell Signaling Technology, Denvers, MA, USA), anti-ROCK1 (1:1000; Cell Signaling Technology),anti-ITGA4 (1:500, Santa Cruz Biotechnology), anti-ITGB7 (1:500, Santa Cruz Biotechnology),anti-gp130 (1:500; Santa Cruz Biotechnology), anti-IL-6 (1:1000; Abcam, Cambridge, UK),phospho-STAT3 (1:1000; Cell Signaling Technology), and anti-GAPDH (1:3000; Abfrontier, Seoul, Korea).The membranes were then incubated with horseradish peroxidase-conjugated secondary anti-mouse IgG (1:5000, Cell Signaling Technology) and anti-rabbit IgG (1: 10,000, Cell SignalingTechnology) for 1 h at room temperature. Bands were detected using a Clarity Western ECL kit (Bio-Rad). Western blotting was performed in triplicate.

### 4.7. Transwell Migration Assay Using Transfection

The migration of naïve and PRL-1(+) PD-MSCs was assessed using a Transwell assay under normoxic and hypoxic (1%) conditions. Naïve and PRL-1(+) PD-MSCs (2 × 10^4^ cells/well) were seeded onto inserts (8µmpore size; Corning, NY, USA) transfected with siPRL-1and miRNA mimic/inhibitor (Integrated DNA Technologies, Coralville, IA, USA)at final concentrations of 50 nM using Lipofectamine 2000 (Invitrogen) for 24 h. The migrated cells in each group were analyzed or fixed with 100% methanol for 10 min and stained with Mayer’s hematoxylin (DAKO). The stained cells in eight randomnon-overlapping fields were counted at a magnification of 200×. The experiments were conducted in triplicate.

### 4.8. Dual Luciferase Assay

To analyze PRL-1, ITGA4, and ITGB7 3’-UTRs containing the binding sites of miR-30a-5p, 340-5p, and 146a-3p, we constructed apmirGLO luciferase miRNA target expression vector (Promega, Madison, WI, USA) and correctly confirmed the results of sequencing (Bioneer, Daejeon, Republic of Korea). Naïve and PRL-1(+) PD-MSCs were seeded on 24-well plates and transfected with each miRNA mimic or negative control (NC) using Lipofectamine 2000 (Invitrogen) for 24 h. The relative luciferase activities were measured by luminescence (Tecan, Switzerland). The primer sequences were as follows: 5’–ctcgagGCACAATACTTGTATA–3’ with *Xho*I site (forward) and 5’–tctagaATATTGGTATGAATGTGG–3’ with *Xba*I site (reverse) for PRL-1; 5’–ctcgagTTCTAACGAGTCCAC–3’ with *Xho*I site (forward) and 5’–tctagaGTAATTTCACTAAGCTC–3’ with *Xba*I site (reverse) for ITGA4; 5’–ctcgagAGGGACACTTACCCAA–3’ with *Xho*I site (forward) and 5’–tctagaGCAGGCATGGGAAGCA–3’ with *Xba*I site (reverse) for ITGB7. The experiment was performed in at least triplicate.

### 4.9. Deep Sequencing and Analysis of Small RNAs

We performed miRNA sequencing experiments on PD-MSCsusing Illumina platforms. We obtained total 10 million clean reads that aligned with the rat genome in the NTx and TTx Naïve groups at one and two weeks in a rat model with BDL and migrated PD-MSCs under normoxic and hypoxic conditions. We compared the normalized counts of mature miRNAs in NTx versus TTx at one week, NTx versus TTx at two weeks, and hypoxic versus normoxic conditions. The construction of small RNA libraries with these samples, deep sequencing, and the analysis of small RNAs were performed by LAS Inc. (Gimpo, Republic of Korea). The expression levels of miRNAs (transcripts per 10 million, TPTM) in the indicated samples were calculated by normalizing the miRNA counts with the total number of clean reads in the small RNA libraries.

### 4.10. Statistical Analysis

The data are expressed as means± standard deviation of at least three independent experiments. Student’s *t*-test was conducted, and *p*< 0.05 was considered statistically significant.

## Figures and Tables

**Figure 1 ijms-20-05299-f001:**
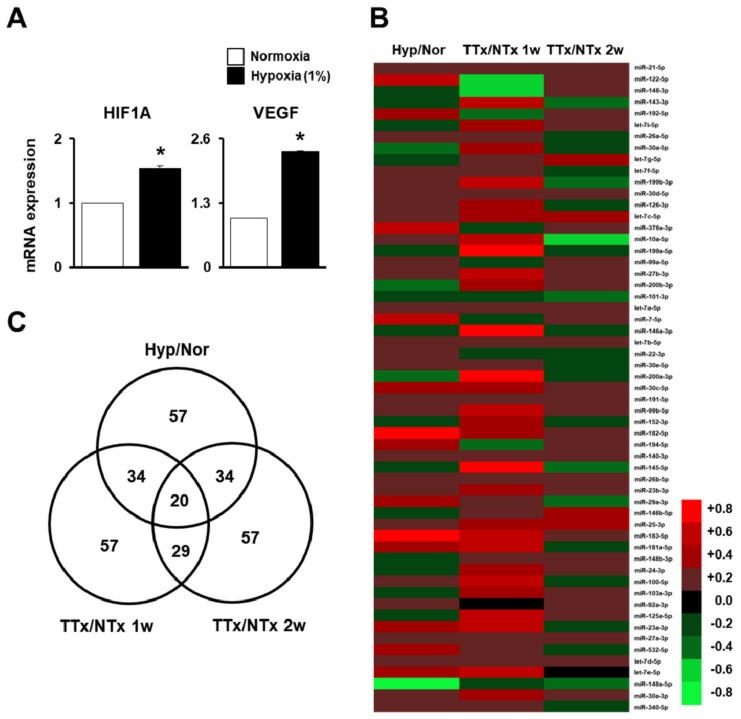
MicroRNA (miRNA) profiling of migrated naïve placenta-derived mesenchymal stem cells (PD-MSCs) under hypoxic conditions and in bile duct ligation (BDL)-injured liver in rats. (**A**) Messenger RNA (mRNA)expression of hypoxia-inducible factor 1 alpha (HIF-1α) and vascular endothelial growth factor (VEGF) in migrated naïve PD-MSCs determined using a Transwell insert system under 1% hypoxic conditions for 24 h, as determinedby quantitative real-time polymerase chain reaction (qRT-PCR). (**B**) Heat map and (**C**) Venn diagram of the microarray results ofmigrated naïve PD-MSCs under hypoxic conditions compared with normoxic conditions (Hyp/Nor), transplanted naïve (TTxNaïve) compared to NTx at one week (TTx/NTx 1w), and TTx Naïve compared to NTx at two weeks (TTx/NTx 2w). qRT-PCR runs were conducted in at least triplicate. Data from each group are shown as means ± SD, determined by Student’s *t*-test; * *p* < 0.05 vs. normoxia.

**Figure 2 ijms-20-05299-f002:**
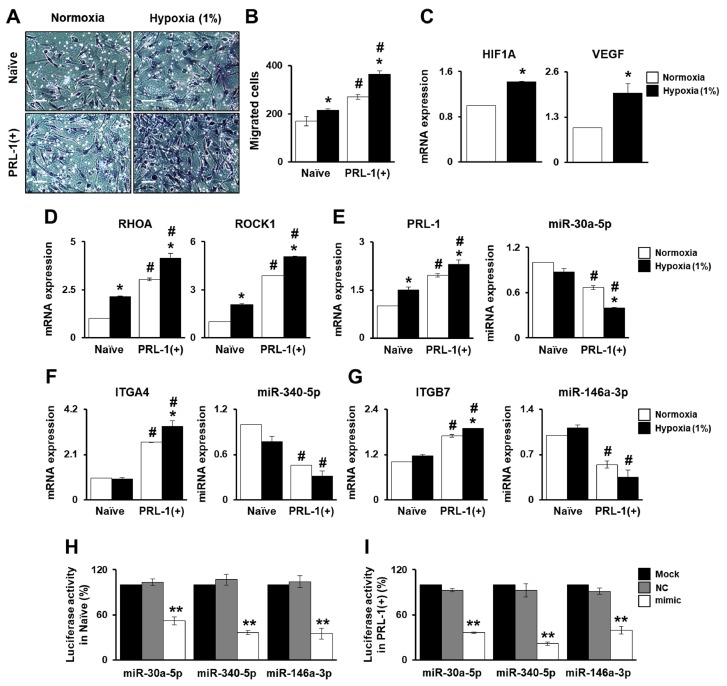
Phosphatase of regenerating liver-1 (PRL-1)-dependent migration ability under hypoxic conditions regulated by miRNAs targeting the integrin family.(**A**) Representative images and (**B**) the number of migrated naïve and PRL-1(+) PD-MSCs using a Transwell insert system under 1% hypoxic conditions for 24 h. (**C**) mRNA expression of hypoxia-inducible factor 1 alpha (HIF-1α) and vascular endothelial growth factor (VEGF) in migrated PRL-1(+) PD-MSCs determined using a Transwell insert system under 1% hypoxic conditions for 24 h, as determinedby qRT-PCR. (**D**) Ras homolog family member A (RHOA) and RHO-associated coiled-coil-containing protein kinase 1 (ROCK1) levelsin migratednaïve and PRL-1(+) PD-MSCs under 1% hypoxic conditions as determined by qRT-PCR. (**E**) mRNA expression levels of PRL-1 and targeted hsa-miR-30a-5p expression, (**F**) Integrin alpha 4 (ITGA4) and targeted hsa-miR-340-5p expression, and (**G**) integrin beta 7 (ITGB7) and targeted hsa-miR-146a-3p expression in migrated PRL-1(+) PD-MSCs under 1% hypoxic conditions for 24 h as determined by qRT-PCR. Luciferase assay of PRL-1, ITGA4, and ITGB7 containing hsa-miR-30a-5p, 340-5p, and 146a-3p binding sitesin (**H**) naïve and (**I**) PRL-1(+) PD-MSCs. Firefly luciferase activities were measured by luminescence. All experiments were conducted in at least triplicate. Data from each group are expressed as means ± SD, determined by Student’s *t*-test. Scale bars = 100 μm.* *p* < 0.05 vs. normoxia, ^#^
*p* < 0.05 vs. naïve, and ** *p* < 0.05 vs. negative control (NC).

**Figure 3 ijms-20-05299-f003:**
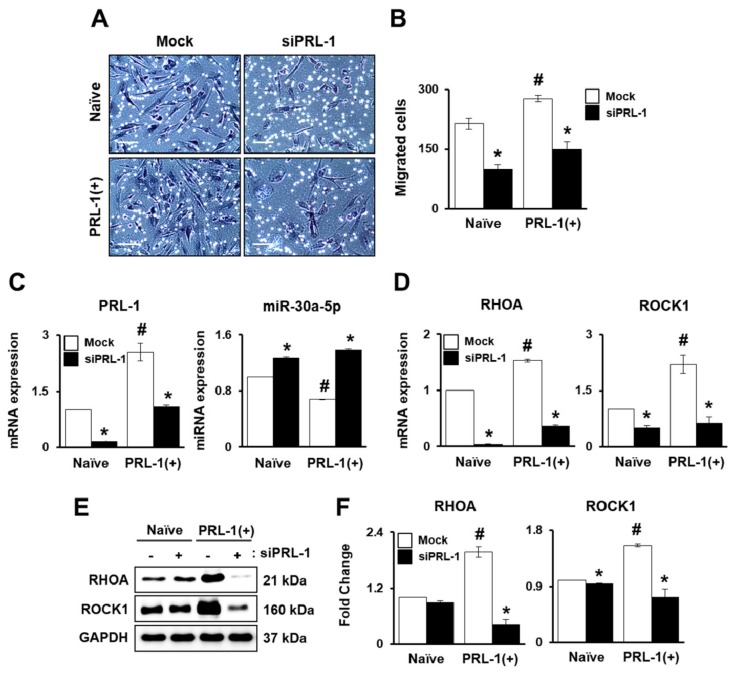
PRL-1-Targeted miRNA expression regulates migration ability through the RHO family. (**A**) Representative images and (**B**) the number of migrated cells in the naïve PD-MSCs and PRL-1(+) PD-MSCs determined using a Transwell insert system following small interfering RNA (siRNA) PRL-1 (siPRL-1) treatment (50 nM) for 24 h. (**C**) mRNA expression of PRL-1 and targeted hsa-miR-30a-5p expression, and (**D**) RHOA and ROCK1 in migratednaïve and PRL-1(+) PD-MSCs determined using a Transwell insert system following siPRL-1 treatment (50 nM) for 24 h as determined by qRT-PCR.(**E**) Protein levels of RHOA and ROCK1 and (**F**) their quantification in migrated naïve and PRL-1(+) PD-MSCs according to siPRL-1 treatment by Western blotting. All experiments were performed in at least triplicate. Data from each group are shown as means ± SD, determined by Student’s *t*-test. Scale bars = 100 μm. * *p* < 0.05 vs. mock and ^#^
*p* < 0.05 vs. naïve.

**Figure 4 ijms-20-05299-f004:**
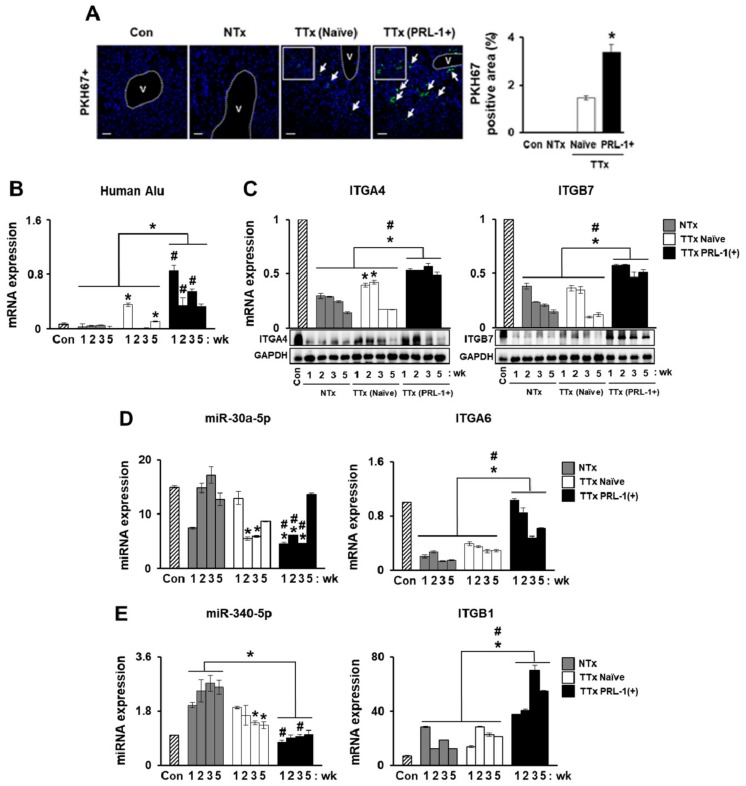
miRNA expression regulates integrin family forPRL-1(+) PD-MSC homingin vivo in a rat model of BDL. (**A**) PKH67 (green)-labeled transplanted cells inBDL-injured rat liver by fluorescence microscopy at one week (arrow = PKH67+ green signals) (blue = 4’,6-diamidino-2-phenylindole; DAPI). mRNA expression levels of (**B**) human-specificAlu sequenceafter the engraftment of naïve (TTxNaïve) and PRL-1(+) PD-MSCs (TTx PRL-1(+)) into injured rat liver compared withthe sham control (Con) and BDL-injurednon-transplantation groups (NTx) at one, two, three, and fiveweeksas determined by qRT-PCR. mRNA and protein levels of (**C**) ITGA4 and ITGB7 and (**D**) integrin alpha 6 (ITGA6)-targeted rno-miR-30a-5p and (**E**) integrin beta 1 (ITGB1)-targeted rno-miR-340-5p expression in rat liver with BDL at one, two, three, and fiveweekspost-transplantation as determined by qRT-PCR.All experiments wereperformed in at least triplicate. Data from each group are shown as means ± SD, determinedby Student’s *t*-test. Scale bars = 100 μm.* *p* < 0.05 vs. NTx and ^#^
*p* < 0.05 vs. TTx Naïve, wk: week.

**Figure 5 ijms-20-05299-f005:**
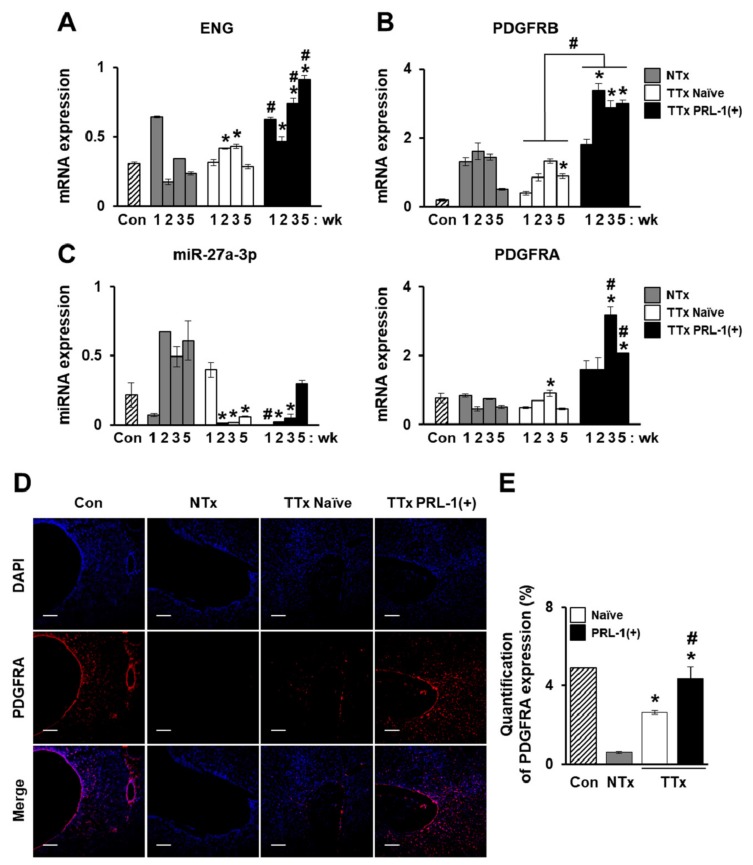
Improved vascular remodeling by PRL-1(+) PD-MSCs through the regulation of miRNA expression by platelet-derived growth factor receptor alpha (PDGFRA)in a BDL-injured rat model. (**A**) qRT-PCR of endoglin (ENG), and (**B**) platelet-derived growth factor receptor beta (PDGFRB) and (**C**) PDGFR alpha (PDGFRA)-targeted rno-miR-27a-3p expression in BDL-injured rat liver tissue after the administration of naïveand PRL-1(+) PD-MSCs at one, two, three, and fiveweeks. (**D**) Localizationof PDGFRA expression and (**E**) quantification of PDGFRA+ cells in rat liver sections from each group (Con, NTx, TTx Naïve, and TTx PRL-1(+)) at one week as determined by immunofluorescence (red, PDGFRA; blue, DAPI). Scale bars = 100 μm. All experiments wereconducted in at least triplicate. Data from each group are expressed as means ± SD, determined by Student’s *t*-test.* *p* < 0.05 vs. NTx and ^#^
*p* < 0.05 vs. TTx Naïve, wk: week.

**Figure 6 ijms-20-05299-f006:**
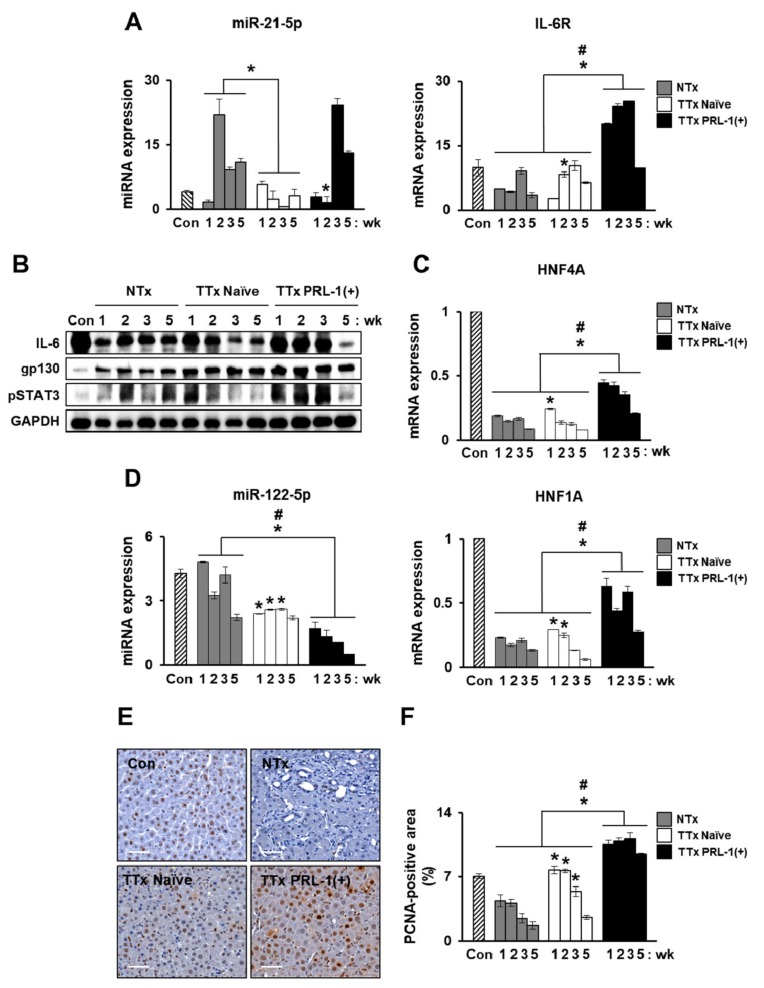
miRNAs mediated hepatic regeneration by PRL-1(+) PD-MSCs in a rat model of BDL through interleukin-6 (IL-6)/signal transducer and activator of transcription 3 (STAT3) signaling. (**A**) qRT-PCR of rno-miR-21-5p-targeted interleukin-6 receptor (IL-6R) and (**B**) Western blotting of glycoprotein 130 (gp130), IL-6, and phosphorylated STAT3 levels in BDL-injured rat liver tissue following the administration of naïve and PRL-1(+) PD-MSCs at one, two, three, and fiveweeks. (**C**) mRNA expression of hepatocyte nuclear factor 4 alpha (HNF4A) and (**D**) miR-122-5p-targeted HNF1 homeobox A (HNF1A) in BDL-injured rat liver tissue by qRT-PCR. (**E**) Proliferating cell nuclear antigen (PCNA) expression in the rat liver sections from each group (Con, NTx, TTxNaïve, and TTxPRL-1(+)) at one week as determined by immunohistochemistry. (**F**) Quantification of the PCNA-positive area in hepatocytes. Scale bars = 50μm. All experiments wereconductedin at least triplicate.Data from each group are expressed as means ± SD, determined by Student’s *t*-test.* *p* < 0.05 vs. NTx and ^#^
*p* < 0.05 vs. TTx Naïve, wk: week.

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
