# Peer review of "Dynamic Regulation of miRNA Expression by Functionally Enhanced Placental Mesenchymal Stem Cells Promotes Hepatic Regeneration in a Rat Model with Bile Duct Ligation"

_ijms, 2019, doi:10.3390/ijms20215299_

Round 1

Reviewer 1 Report

I am happy with the current revision. It is suitable for publication in IJMS.

Author Response

Manuscript ID: ijms-623184

Reviewer #1:

General comments: I am happy with the current revision. It is suitable for publication in IJMS.

Author’s reply: We greatly appreciate the reviewer’s positive statement that "I am happy with the current revision. It is suitable for publication in IJMS."

Reviewer 2 Report

no further comment or modification would be required from my side.

Author Response

Manuscript ID: ijms-623184

Reviewer #2:

General comments: no further comment or modification would be required from my side.

Author’s reply: We are grateful for the reviewer’s positive statement that "no further comment or modification would be required from my side."

Reviewer 3 Report

The publication covers an interesting topic from the scientific and application point of view.
The introduction is very synthetically developed. The Material and Methods chapter is elaborated in detail. Test results are documented with charts and photos. Figure 4A should be enlarged because it is hard to read.
You should also make a language correction because the style and grammar are incorrect and many sentences are incomprehensible to the reader.

Author Response

Manuscript ID: ijms-623184

Reviewer #3:

General comments: The publication covers an interesting topic from the scientific and application point of view. The introduction is very synthetically developed. The Material and Methods chapter is elaborated in detail. Test results are documented with charts and photos. Figure 4A should be enlarged because it is hard to read. You should also make a language correction because the style and grammar are incorrect and many sentences are incomprehensible to the reader.

Author’s reply: We greatly appreciate the reviewer’s positive statement that "The publication covers an interesting topic from the scientific and application point of view. The introduction is very synthetically developed. The Material and Methods chapter is elaborated in detail. Test results are documented with charts and photos." As reviewer mentioned, Figure 4A and some sentences are corrected in our revised manuscript. Also, it was corrected by American Journal Experts (AJE) and their verification code is 566C-2BF0-4BB1-436E-576P.

This manuscript is a resubmission of an earlier submission. The following is a list of the peer review reports and author responses from that submission.

Round 1

Reviewer 1 Report

The manuscript by Kim JY et al. addressed the possible therapeutic effect of human placenta-derived mesenchymal stem cells (PD-MSCs), as well as those with an exogenously-introduced phosphatase of regenerating liver-1 (PRL-1) gene, on hepatic regeneration upon transplantation to a rat model with bile duct ligation-induced liver injury, with a particular focus on the possible mechanistic role of microRNAs (miRNAs). While the study is of potential interest, the set of analyses executed in the present manuscript are superficial and still immature, and do not sufficiently provide solid evidence for the effect of PD-MSCs nor functional significance of miRNAs. In particular, the study on miRNAs was solely limited to their expression profiles and no functional analysis (i.e., gain-of-function and loss-of-function experiments) was done. Several other concerns and questions should be addressed to clarify and strengthen the authors claims and conclusions, as follows:

The nature of the cell/tissue samples subjected to the miRNA profiling was not clearly described. What did the authors mean by “invaded” naïve PD-MSCs under hypoxic conditions (page 3, line 94)? Did they correspond to the cells “migrated and invaded to endothelial cells” as mentioned in Introduction, or those simply “migrated” in the Transwell migration assay described in Materials and Methods? If in the former case, did the authors use the specific PD-MSCs population separated from the endothelial cells, or the mixture of these different types of cells?

With regard to the liver samples of rats with BDL administered naïve PD-MSCs, did the authors prepare PD-MSC-specific (or enriched) population from the livers for analyses, or just simply use the dissected liver tissue samples as they were (without any cell isolation/enrichment procedure). If in the latter case, how much extent among the total cell populations did the transplanted PD-MSCs occupy in those liver samples? The authors should also describe whether or not the miRNA profiling procedure was able to discriminate miRNAs of human origin from those of rat origin. If not, the miRNA profiling data obtained from the transplanted liver samples could represent those in the host liver tissue, rather than those in the transplanted PD-MSCs.

It seems that the authors were often confused about the regulatory relationship between miRNAs and their target genes: in general, the former regulate the expression of the latter with cognate target sequences in their UTR, but not vice versa. For instance, the target sequence prediction shown in Table 3 suggested that rno-miR-340-5p and rno-miR-30a-5p regulated expression of ITGB1 and ITGA6. The notion that “the integrin family regulates miRNA expression for … “ (page 7, line 200) should thus be opposite, nor was it supported by any experimental evidence.

Engraftment and pro-regenerative role of the transplanted PD-MSCs were not sufficiently characterized and should be supported by more convincing data. For engraftment, only the expression of human ALU sequence was analyzed (Figure 4A). Histological analyses should be performed to clearly demonstrate whether, where, and for how long the transplanted PD-MSCs resided in the recipient rat livers. Quantitative data regarding the extent of engraftment should also be provided. It is not clear whether the increased cell proliferation in the transplanted (TTx) animals indeed correlated with improved liver regeneration. How about the liver-to-body weight ratio? Did the transplantation ameliorate inflammatory and fibrotic responses? Serum markers for cholestasis (e.g., bilirubin and bile acids) and hepatocyte injury (e.g., AST and ALT) should be analyzed.

How many different clones/batches of PRL-1(+) PD-MSCs were analyzed in the study to confirm the reproducibility of the results? Did the PRL-1 gene used to establish the PRL-1(+) PD-MSCs contain the 3’-UTR sequence including the hsa-miR-30a-5p target site?

Page 3, line 108, “the miRNA profiles of naïve PD-MSCs with migration ability under hypoxic conditions and of BDL-injured rat liver samples were identical”. The profiles were not identical, albeit overlapping.

Page 5, line 132, “The invaded naïve PD-MSCs showed decreased ITGA4 expression under hypoxic conditions compared with normoxic conditions.” The ITGA4 expression did not change according to Figure 2C.

Page 5, lines 139–142. As mentioned above, there was no direct evidence showing that hsa-miR-146a-3p was capable of downregulating the expression of ITGB7. No evidence supporting the notion that the integrin family was functionally involved in miRNA regulation was provided either.

Page 5, line 153, “we found that PRL-1(+) PD-MSCs had increased migration ability under hypoxic conditions compared with normoxic conditions”. No corresponding data or literature can be found in the manuscript.

Figure 3, panels C and D. In the naïve PD-MSCs, the siRNA-mediated knockdown of PRL-1 resulted in complete suppression of the RHOA expression (panel D), while it induce only a marginal (around 1.3-fold) increase in the miR-30a-5p expression. Can overexpression of miR-30a-5p alone sufficiently suppress the RHOA expression?

Figure 4, B–D. The species (i.e., human or rat) for each genes analyzed should be clarified. It should be reasonable to assume that ITGA4 and ITGB7 in panel B were of human origin, and if so, the extremely high levels of their expression in the sham control (Con) animals indicate that the experiments were not properly done. It is also hard to understand why expression of ITGA6 in panel B was high in Con and decreased in the NTx and TTx Naïve groups, even if the gene was of rat origin.

Figure 5D. Images at lower magnification should also be provided, with the positions of the portal and central veins being clearly indicated. It seems unusual that there was no expression of PDGFRA observed in the NTx liver. PDGFRA should be expressed in mesenchymal cells such as hepatic stellate cells in the liver (Kikuchi A and Monga SP. Gene Expr. 2015;16(3):109-27. PMID: 25700367), so that the nature of vascular remodeling should be confirmed by using other specific markers for liver sinusoidal endothelial cells.

Figure 6. In panel A, gene expression of the IL-6 ligand should also be analyzed. In panel B, the IL-6 protein was highly expressed in the Con liver (which itself seems quite strange), while there was little activation/phosphorylation of STAT3 therein. It is also difficult to understand why the levels of HNF4A and HNF1A gene expression (in panel C), as well as the PCNA+ proliferating cells (in panels E and F), were extremely high in the Con animals. Overall, these results cast doubt on the validity, reproducibility and quantitativity of the presented data.

Author Response

August 2, 2019

Cover letter

Article Submission in International Journal of Molecular

Dear Editor,

We greatly appreciate your careful evaluation of our manuscript (IJMS-550864) entitled: “Dynamic regulation of miRNA expression by functionally enhanced placental mesenchymal cells promotes hepatic regeneration in a rat model with bile duct ligation” We also thank you for your patience because our revision took somewhat more time to address all issues raised by the reviewers.

We were really encouraged by the reviewers’ positive comments and constructive suggestions. I am happy to report that we have successfully addressed all issues and concerns through additional data and subsequent revision of our manuscript, as detailed in the following response page. Changes are highlighted in red in the revised manuscript,.

In summary, based on the insightful and constructive criticisms provided by both referees, we believe that our manuscript is significantly improved and we hope that you will consider it suitable for publication in International Journal of Molecular Sciences.

The material contained herein has not been published previously by any of the authors, and is not under consideration for publication in another journal at this time.

Very sincerely yours,

Gi Jin Kim, Ph.D.

Associate Professor

Department of Biomedical Science, CHA University

689, Sampyeong-dong, Bundang-gu, Seongnam-si, Gyeonggi-do, Republic of Korea.

Tel: 82-31-881-3687, Fax: 82-31-881-4102, e-mail: [email protected]

Reviewer 2 Report

In this manuscript, using an elegant approach Kim et al elaborately investigated the miRNA expression profiling of PD-MSCs in promoting hepatic regeneration in BDL-rat model. The authors’ efforts worth appreciation given the amount of work they have put in. The experiments that they have presented are extensive and nicely controlled. However, I have some minor concerns, which should be addressed, before this could be recommended for publication in “IJMS”.

Comments

1.    The whole manuscript is written in a bit complicated way, which could be hard to follow for readers outside the field. Many sentences are very long. As for example, in line 80, “Moreover, umbilical cord MSCs (UC-MSCs) transplantation reduced the severity of hepatic fibrosis in a CCl4 -injured mouse model through the suppression of miR-199 expression targeted to the 3'-UTR site of keratinocyte growth factor mRNA to decrease translation.” This type of sentences should be simplified to easily convey the message.

2.    In line 108, the authors concluded that “The data suggested that the miRNA profiles of naïve PD-MSCs with migration ability under hypoxic conditions and of BDL-injured rat liver samples were identical”. I don’t think they can conclude this based on how they have presented the data before. To show miRNA expression similarity, they need to mention out of how many, certain number of miRNAs show identical expression patterns. A percentage representation would be preferred here, followed by examples of specific miRNAs.

3.    In the discussion section, can they add a paragraph in details about a plausible mechanism of how does PRL-1 control the expression of has-mir-30a-5p. In fact, a model figure about this mechanism comparing different conditions (e.g. hypoxic vs. normoxic) would be immensely helpful. I think this is a main message of the manuscript, and such a message has to be clearly conveyed.

Author Response

In this manuscript, using an elegant approach Kim et al elaborately investigated the miRNA expression profiling of PD-MSCs in promoting hepatic regeneration in BDL-rat model. The authors’ efforts worth appreciation given the amount of work they have put in. The experiments that they have presented are extensive and nicely controlled. However, I have some minor concerns, which should be addressed, before this could be recommended for publication in “IJMS”.

Author’s reply: We greatly appreciate the reviewer’s positive statement that "The authors’ efforts worth appreciation given the amount of work they have put in. And they have presented are extensive and nicely controlled”.

The whole manuscript is written in a bit complicated way, which could be hard to follow for readers outside the field. Many sentences are very long. As for example, in line 80, “Moreover, umbilical cord MSCs (UC-MSCs) transplantation reduced the severity of hepatic fibrosis in a CCl4 -injured mouse model through the suppression of miR-199 expression targeted to the 3'-UTR site of keratinocyte growth factor mRNA to decrease translation.” This type of sentences should be simplified to easily convey the message.

Author’s reply: Thank you for your encouragement comment. As you commented, these are simply corrected in the revised manuscript. Corrected parts are marked in red.

In line 108, the authors concluded that “The data suggested that the miRNA profiles of naïve PD-MSCs with migration ability under hypoxic conditions and of BDL-injured rat liver samples were identical”. I don’t think they can conclude this based on how they have presented the data before. To show miRNA expression similarity, they need to mention out of how many, certain number of miRNAs show identical expression patterns. A percentage representation would be preferred here, followed by examples of specific miRNAs.

Author’s reply: Thank you for your comment. To address that point, please refer to our response to Point #2 and Point #6 of reviewer1.

In the discussion section, can they add a paragraph in details about a plausible mechanism of how does PRL-1 control the expression of has-mir-30a-5p. In fact, a model figure about this mechanism comparing different conditions (e.g. hypoxic vs. normoxic) would be immensely helpful. I think this is a main message of the manuscript, and such a message has to be clearly conveyed.

Author’s reply: We greatly appreciate the reviewer’s thoughtful comments. These are corrected in the revised manuscript. Corrected parts are marked in red.

Reviewer 3 Report

Manuscript ID: ijms-550864

The authors investigate the mechanistic effect of placenta-derived mesenchymal stem cells in the context of a rat model with hepatic failure with bile duct ligation. Although a range of technique is used to investigate mechanistic processes, a lot is done on mRNA level and not confirmed at the protein level. Also some more specific inhibitory experiments are required to really draw solid conclusions.

General comment

For every figure, please provide in the legend the number of replicates or experiments, as well as the statistical test used.

3 individuals unfortunately do not constitute a solid set of data to perform statistical tests and provide convincing evidences for the readers. Although some experiments are difficult to replicate, for some it might be possible to increase the number of replicate per group.

Figure 1

For the heatmap, please provide a legend for the scale so that the reader can interpret it, from -0,8 to 0,8 , is it Z-score ? or relative expression ? , please clarify.

Line 109 “The data suggested that the miRNA profiles of naïve PD-MSCs with migration ability under hypoxic conditions and of BDL-injured rat liver samples were identical.”

From the data displayed with the venn diagram and the heatmap, the miRNA profile of these two “samples” are not really identical, could the authors display or show evidences more straight forward to interpret, or clarify?

Figure 2

Since mRNA don’t equal to functional protein expression at the surface of cells, some of the main findings could be confirmed by Flow cytometry using fluorescent antibodies, such as ITGA4 and ITGB7 for instance.

Same comment is valid for figure 4

Figure 3

Figure 3. Is entitled “PRL-1-Targeted miRNA expression regulates migration ability through the RHO family “ , however although RHO mRA expression does vary, its doesn’t show that these are directly involved in the migration (and not only co-vary). Migration experiments with specific inhibitors could be done to ensure such effect and conclusions for instance.

Figure 5.

Figure 5. is entitled  “Improved vascular remodeling by PRL-1(+) PD-MSCs (…)” thus in this context:

In order to show vascular remodeling, please provide some staining for vasculature with immunostaining classically accepted and used such as CD31 for instance. In addition, to speculate on vascular remodeling, some quantitative imaging analysis should be done comparing two time points.

Figure 6

The authors speculate that PRL-1(+) PD-MSCs effect is done through p-STAT3, however, a high p-STAT3 is seen in the western blot of TTx naïve individuals, could the authors comment on that ?

Author Response

(The authors gave the same response as above.)

Reviewer 4 Report

Publication of a high scientific aspect and possible future clinical use. Very accurately described methods allow you to repeat the experiment.

The results are in graphical form, that is, legible charts. The sizes of some figures (5D) could be slightly larger, which would increase their readability.
Some items of the quoted literature I would propose to change into more recent, e.g. 6,15,16,50.
The publication is very valuable in general.

Author Response

(The authors gave the same response as above.)

Round 2

Reviewer 3 Report

No additional comments.